# The Effect of Potassium–Nitrogen Balance on the Yield and Quality of Strawberries Grown under Soilless Conditions

Amal Nakro [1], Ahmed Bamouh [1], Hajar Bouslama [1], Alberto San Bautista [2,*] and Lamiae Ghaouti [1]

1  Department of Plant Production, Protection and Biotechnology, Hassan II Institute of Agronomy and Veterinary Sciences, BP 6446 Madinate Al Irfane, Rabat 10112, Morocco
2  School of Agricultural and Environmental Engineering, Polytechnic University of Valencia, 46006 Valencia, Spain
*  Correspondence: asanbau@prv.upv.es

**Abstract:** This research aims to evaluate the effect of the potassium–nitrogen balance on strawberry productivity and quality parameters in soilless conditions. A trial was conducted at the Agronomic and Veterinary Hassan II Institute glasshouse in Rabat, during the 2018–2019 crop year. The pot experiment began on November 13 using three different strawberry cultivars: Fortuna, San Andreas and Sabrina. Three nutrient solutions were tested by increasing the potassium–nitrogen balance during the growth stage and decreasing it during the fruit-production stage: S1 (1.3/2.0), S2 (2.6/1.0) and S3 (3.0/0.6). For all treatments, the total dose of fertilizing elements N, P, K, Ca and Mg was identical. The experimental design was a randomized complete block with three replications. The results show that strawberry plants receiving a nutrient solution with a high K:N balance during the growth period and a low balance during the production period present the higher growth and fruit levels. Moreover, the nutrient solution with the 2.6/1.0 balance significantly increased the chlorophyll index by 8%, yield by 30% (7.9 t ha$^{-1}$), total soluble solids and dry matter content by 14% and 15%, respectively, and improved taste and fruit shelf-life by 10% and 19%, respectively.

**Keywords:** fertigation; nutrient solution; ratio; productivity; total soluble solids; maturity index



## 1. Introduction

In Morocco, strawberry cultivation occupies a prominent position in the berry sector with 29% of the total area cultivated with red berries, this represents 50% of Moroccan berry exports [1].

In the early 1990s, as a result of good production results and the increasing interest of European markets in Moroccan strawberries, the cultivated area and production practices have experienced a significant evolution, shifting from 750 ha with 31,000 t harvested in 1995 to 3400 ha during the 2020–2021 crop year, with an average production of 102,000 t [1].

Given the strong competition in the strawberry export market, Moroccan producers are opting for high-performance varieties in terms of production strategies and physico-chemical qualities. To achieve this, growers must master appropriate techniques, mainly plant nutrition, in order to balance productivity, regularity of the production, and the quality of strawberry fruits during this cycle. The marketing of fresh strawberries is essentially conditioned by their quality [2], for which total soluble solids (TSS), acidity and the fruit's visual appearance are the main determinants [3].

Strawberry growers will gain higher productivity rates when cultivating strawberries with the incorporation of proper plant nutrition management. More specially, the nutritional elements of potassium and nitrogen are of primary importance to strawberry farmers, as these macronutrients are the most highly absorbed by strawberries [4] and have a significant impact on their yield and quality [5,6]. Several studies have shown that nitrogen nutrition plays an essential role in fruit growth, development and production processes [7].

Potassium is involved in several metabolic processes such as photosynthesis, protein synthesis and enzymatic activities [8]. Rodas et al. [9] showed that quality parameters such as total soluble solids, titratable acidity and fruit juice pH are influenced by combined doses of potassium and nitrogen. Numerous studies conducted on strawberries to search for the optimum K:N balance reported optimal values of 1.64 [10], 1.54 [11,12], 2.05 [4], 1.72 [9] and 2.60 [13]. These values are higher than the K:N balance adopted by Moroccan strawberry farmers, which is 1.2 during the strawberry plant's growth period. Therefore, the concept of potassium–nitrogen balance is crucial to the successful management of strawberry mineral nutrition.

Plant nutrition management in strawberry cultivation is more effective and accurate in hydroponic systems [14]. Defined as the cultivation of plant species in an environment isolated from soil [15], soilless or substrate cultures provide plants with a growing environment free of root diseases and insects [16] and prevents the use of fumigants and herbicides while reducing environmental pollution levels. The disadvantages of conventional strawberry production include root diseases such as verticillium wilt, which persist in the soil from year to year.

Soilless strawberry farming requires a high level of skill and expertise in crop and irrigation management techniques [16]. Owing to less rhizosphere volume in soilless conditions, strawberry plants empty the soil water supply more rapidly in soilless compared to the field conditions [17]. Water and fertilizer management requires tailored practices to maintain optimal crop conditions [18]. Soilless strawberries require 1.5 to 2.5 L of substrate per plant [7], which should be selected depending on several criteria, aeration, water-holding capacity, chemical properties [19], and strawberry variety planted [20–22].

The objective of our research is to study the effect of the potassium–nitrogen balance on the productivity and quality of strawberry plant fruits in soilless growing conditions, while generating a strawberry nutrition program with an optimal potassium–nitrogen balance that improves both fruit yield and quality.

## 2. Materials and Methods

A pot experiment was conducted in the greenhouse of the Hassan II Agronomic and Veterinary Institute in Rabat, Morocco (33°58′43.00″ N 6°51′50.75″ O; altitude: 127 m). Strawberry cultivars Fortuna, San Andreas and Sabrina were transplanted into 12 L bags (length = 60 cm, width = 20 cm, depth = 15 cm) on 13 November 2018, with the plants placed 25 cm apart in each bag. The substrate used was coarse river sand with the following characteristics: pH: 6.6, EC: 0.85 dS m$^{-1}$, total porosity: 27% and water content at field capacity: 15%. The plants were irrigated with three nutrient solutions with three different K:N values (Table 1). The amount of solution applied (50–500 mL per plant per day) was adjusted to transpiration to ensure 20% drainage [23,24]. The bags were equipped with fertigation tanks, drip irrigation systems and drains with taps to collect the drainage and allow for fertigation management. Furthermore, the pH and EC levels of the nutrient solutions and drainage were measured in order to meet the specific requirements of the strawberry plants: EC: 1–2 dS m$^{-1}$ and pH: 5.5–6.5 [25,26]. The mean air temperature at the trial site varied from an initial value in the range 15 °C–19 °C at the beginning of the experiment (mid-November), reaching temperatures up to 24 °C by mid-June. The maximum air temperature ranged from 17 to 20 °C during the growth period and from 19 to 26 °C during the production period. Minimum temperatures during these stages were 8 and 12 °C, respectively.

Three nutrient solutions (S1, S2 and S3) were combined with three cultivars following a randomized complete block design, with 3 replications and 12 plants per replication. The three strawberry cultivars widely planted in the Loukkos region [27], proportionally 31.8% for Sabrina, 14.6% for San Andreas and 14.2% for Fortuna, represent the principal varieties imported by Morocco in 2017 [28]. We tested three nutrient solutions, increasing the potassium–nitrogen balance during the growth period (November–January) and then decreasing it during the fruit-production period (February–June):

— S1: Control nutrient solution, maintaining it at a low-balance (1.3) level during the growth period and at a high-balance (2.0) level during the production period. This is a common practice of Moroccan strawberry farmers [29].
— S2: High-balance nutrient solution (2.6) during the growth period, and low-balance (1.0) during the production period.
— S3: very-high-balance nutrient solution (3.0) during the growth period, and very-low-balance (0.6) during the production period.

**Table 1.** Nutrient solutions composition (mmol $L^{-1}$).

| Stage | Nutrient Solution | K:N Balance | N-NO$_3$ | N-NH$_4$ | P | K | Ca | Mg | SO$_4$ | pH | EC (dS m$^{-1}$) |
|---|---|---|---|---|---|---|---|---|---|---|---|
| Growth | S1 | 1.3 | 5.94 | 1.63 | 1.52 | 3.82 | 3.40 | 2.47 | 1.31 | 6.67 | 2.28 |
| | S2 | 2.6 | 5.94 | 1.63 | 1.52 | 7.41 | 3.40 | 2.47 | 2.55 | 6.61 | 2.77 |
| | S3 | 3.0 | 5.94 | 1.63 | 1.52 | 8.80 | 3.40 | 2.47 | 3.03 | 6.62 | 2.53 |
| Production | S1 | 2.0 | 5.35 | 0.75 | 0.76 | 4.44 | 3.65 | 1.74 | 1.53 | 6.26 | 1.60 |
| | S2 | 1.0 | 5.35 | 0.75 | 0.76 | 2.29 | 3.65 | 1.74 | 0.79 | 6.25 | 1.77 |
| | S3 | 0.6 | 5.35 | 0.75 | 0.76 | 1.46 | 3.65 | 1.74 | 0.50 | 6.35 | 1.70 |

### 2.1. Growth and Production Parameters

In order to evaluate the effect of nutrient solution with different balances on the parameters of growth, yield and quality of strawberry plants, the fruits were picked every week during the crop season, from February to June.

The growth parameters recorded were the chlorophyll index measured by a hand-held chlorophyll meter (CCM-200 plus Opti-Sciences, USA), and stomatal conductance measured by a leaf porometer (AP4 from Delta-T Devices, UK). Two mature leaves per plant of each replication (nutrient solution x variety) and three determinations per leaf were sampled on 14 March 2019; the measurements were obtained at 12:00 p.m.

The pomological analyses performed were for number of fruits per plant and average fruit-weight harvested, early (31 March 2019) and total (30 June 2019) fruit yield values were measured through the cumulative weight of strawberries produced during the entire harvest period, which lasted for more than 4 months (February–June), and the fruit size was measured with a digital caliper. Fruits obtained from each plant of each replication (nutrient solution x variety) were sampled twice a week; the fruits were harvested at 9:00 a.m. Immediately following the harvest, the fruits were weighed.

The quality parameters concerned were total soluble solids (TSS) measured by using a digital refractometer (HI 96801, HANNA), and titratable acidity (TA) determined by the titration method for fruit juice [30]. The titrated volume of 0.1 M NaOH was recorded using a benchtop pH meter (CONSORT P107) to an endpoint of pH 8.1, and the acidity was calculated as a percentage of the citric acid content. The maturity index was calculated as the TSS/TA ratio.

The organoleptic properties of the fruits were determined by sensory analysis including the global taste of strawberries through a tasting workshop involving 12 people. The shelf-life of the strawberries was determined by the percentage of the fruits' weight loss after 5 days of storage for a sample of 10 strawberries in plastic trays with perforated lids, kept in a refrigerator at 3 °C. After obtaining the fresh weight of the sampled strawberry fruits, the fruits were placed in an oven for 48 h at 80 °C and then weighed for their dry weight [30] in order to measure the dry matter content of strawberry fruit.

### 2.2. Statistical Analysis

The data were analyzed for significant differences using Statgraphics Centurion statistical software. Analysis of variance (ANOVA 2) and comparison of means by least significant difference test ($p < 0.05$), when the *F*-ratio was statistically significant ($p < 0.05$), were performed for each parameter studied in order to evaluate the statistical significance of the tests.

## 3. Results and Discussion

### 3.1. Effect of K:N Balance on the Growth Parameters of Strawberry Plants Grown in Soilless Conditions

Both factors, cultivar and nutrient solution, were statistically significant ($p < 0.01$) in the chlorophyll index; however, the cultivar factor was highly significant and accounted for 63.8% of the total variability, while the nutrient solution factor accounted for 13.8%. In addition, the cultivar–nutrient solution interaction was statistically significant ($p < 0.05$), accounting for 9.6%. In relation to stomatal conductance, no statistically significant influence for any of the studied factors or interaction was detected in the experiment. The plants of the Fortuna cultivar presented a higher chlorophyll content with statistically significant differences, with respect to both San Andreas ($p < 0.05$) and Sabrina varieties.

Chlorophyll levels measured at 121 days after planting showed that there was a significant difference between the three nutrient solutions. Nutrient solution S2 with a high K:N balance during the growth period recorded the highest leaf chlorophyll index (Table 2 and Figure 1). The highest leaf stomatal resistance was also recorded in plants receiving the S2 solution. However, these differences were not statistically significant (Table 2).

**Table 2.** Mean values, standard error ($\pm$) and analysis of variance of chlorophyll index (CI) and stomatal conductance (s cm$^{-1}$) for three strawberry cultivars (Fortuna, San Andreas and Sabrina) irrigated with three nutrient solutions.

| Factor | Chlorophyll Index (CI) | Stomatal Conductance (s cm$^{-1}$) |
|---|---|---|
| Cultivar | | |
| Fortuna | 48.3 $^z$ a $\pm$ 2.8 $^y$ a | 0.84 |
| San Andreas | 46.1 $\pm$ 2.8 b | 0.69 |
| Sabrina | 39.1 $\pm$ 2.9 c | 0.64 |
| LSD ($p < 0.05$) | $\pm$2.14 | - |
| Nutrient solution | | |
| S1 | 43.3 $\pm$ 4.6 b | 0.76 |
| S2 | 47.1 $\pm$ 3.8 a | 0.78 |
| S3 | 43.1 $\pm$ 3.0 b | 0.62 |
| LSD ($p < 0.05$) | $\pm$2.14 | - |
| ANOVA (df) | | |
| Factor (df) | Percentage of the total sum of squares | |
| Cultivar (2) | 63.8 ** | 5.00 $^{n.s.}$ |
| Nutrient solution (2) | 13.8 ** | 3.46 $^{n.s.}$ |
| Cultivar x nutrient solution (4) | 9.6 * | 7.62 $^{n.s.}$ |

K:N balance: S1 (1.3/2.0), S2 (2.6/1.0), S3 (3.0/0.6) in growth and productive period. $^z$: mean value; $^y$: standard deviation. Values within a column with different letters differ at $p < 0.05$ using the LSD test. * and ** indicate significances at $p < 0.05$ and $p < 0.01$, respectively; $^{n.s.}$: non-significant

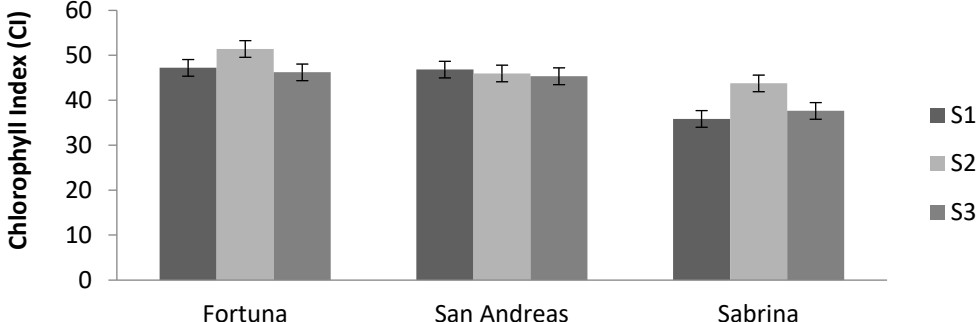

**Figure 1.** Mean values of chlorophyll index measured for three strawberry cultivars (Fortuna, San Andreas and Sabrina) irrigated with three nutrient solutions. K:N balance: S1 (1.3/2.0), S2 (2.6/1.0), S3 (3.0/0.6) in growth and productive period. Vertical bars indicate LSD value ($p < 0.05$).

Since the double interaction is statistically significant ($p < 0.05$), it is necessary to study its effect on the chlorophyll content of the plants (Figure 1). It was observed that in the cultivars Sabrina and Fortuna the highest chlorophyll content was present in the plants irrigated with the S2 nutrient solution, while in the San Andreas cultivar, there were no statistically significant differences between the nutrient solutions. The highest chlorophyll content for the Sabrina cultivar was obtained in the plants irrigated with the S2 solution, with statistically significant differences present relative to the other two cultivars. No statistically significant differences were detected between the Fortuna and San Andreas varieties for the chlorophyll content in plants irrigated with S1 and S3, while in S2 the highest chlorophyll content was obtained for the Fortuna cultivar.

Strawberry plants that received higher potassium quantities during the growth period presented high leaf chlorophyll index and stomatal resistance properties. These results are in agreement with those obtained by Maynard [31], Babicz [32], Hamano et al. [33] and Yagmur et al. [34], who show in their studies that potassium promotes the uptake of nutrients by plants and increases the photosynthesis rate, thus emphasizing the leaves' chlorophyll content due to the strong relationship between the two growth parameters [35]. Owing to the high demand for nitrogen during the growth period, there is also an increased demand for potassium as it is involved in several plant processes such as photosynthesis, protein synthesis, enzyme activities and the regulation of stomata function [36] and efficient nitrogen use [37,38]. Sarıdaş et al. [39] showed in their research that leaf nitrogen content was positively correlated to potassium levels. Indeed, Lieten and Misotten [12] and Tagliavini et al. [10] showed that during the growth period up to the green fruit stage, potassium uptake is greater than that of nitrogen. The positive effect of nutrient solution on the growth parameters of the plants was also observed in a strawberry trial where Nakro et al. [40] reported that the application of a nutrient solution with a high K:N balance during the growth period revealed the highest leaf chlorophyll content for the three cultivars studied. In the same manner, it was also observed that potassium fertilization increased the chlorophyll levels in strawberry, raspberry and blueberry plants [41].

Yagmur et al. [34] and Szczerba et al. [42] reported in their studies that plants depend on potassium for the regulation of stomatal opening and closing processes, and also showed that potassium is involved in the transport of sugars and other nutrients through the xylem.

*3.2. Effect of K:N Balance on the Productivity Parameters of Strawberry Plants Grown in Soilless Conditions*

The variability of the factors studied ranged from 0.9 to 83.1% of the total variability. From these results, it can be observed that the high statistically significant influence ($p < 0.01$) of the cultivar factor on the fruit size stands out, both in the early and total production stages, presenting values of 83.1 and 72.8%; on the other hand, the nutrient solution factor is not statistically significant (0.9 and 4.6%, respectively). The same occurs for the early and total yield parameters, presenting values between 57.6 and 63.06% ($p < 0.01$); for this parameter, however, the nutritive solution exhibited a statistically significant effect ($p < 0.01$) with variability values of 27.7 and 13.3% for the early and total yields, respectively. This greater statistical influence corresponds to the strong influence of the nutrient solution factor on the mean fruit weight ($p < 0.01$), since the variability explained by this factor presented values of 40.6 and 53.3% in the early and total production stages, respectively. In relation to the double interactions that occurred, they were statistically significant ($p < 0.05$) only for the early yield and fruit size obtained during the total production process.

The highest early yield value was obtained for the Sabrina cultivar, with statistically significant differences ($p < 0.05$) relative to San Andreas; the lowest early yield value was obtained for Fortuna plants. The same differences between cultivars were observed for fruit size. In relation to the total yield, plants of the cultivar San Andreas presented the highest value, with statistically significant differences relative to Sabrina, and the cultivar Fortuna was the least productive cultivar.

The analysis of variance showed that the yield was significantly affected by the potassium–nitrogen balance. The recorded values for the production parameters reveal that the largest fruit size and highest yield were presented by plants that received a higher potassium dose during the growth period (S2). However, the differences observed between the three solutions for fruit size were not statistically significant.

Table 3 shows that the S2 nutrient solution increased the fruit size and yield by 2% and 30% (7.9 t ha$^{-1}$), respectively, compared to the S1 solution, which represents the practice of strawberry growers.

**Table 3.** Mean values, standard error ($\pm$) and analysis of variance of productivity parameters (fruit size, weight and yield) for early (31 March) and total production (30 June) for three strawberry cultivars (Fortuna, San Andreas and Sabrina) irrigated with three nutrient solutions.

| Factor | Early | | | Total | | |
|---|---|---|---|---|---|---|
| | Fruit Size (mm) | Fruit Weight (g) | Yield (t ha$^{-1}$) | Fruit Size (mm) | Fruit Weight (g) | Yield (t ha$^{-1}$) |
| Cultivar | | | | | | |
| Fortuna | 22.1 $^z$ $\pm$ 2.4 $^y$ c | 16.4 | 2.49 $^z$ $\pm$ 1.1 $^y$ c | 25.4 | 17.1 | 11.4 $^z$ $\pm$ 8.8 $^y$ b |
| San Andreas | 27.1 $\pm$ 2.9 b | 18.4 | 4.42 $\pm$ 1.2 b | 29.9 | 18.5 | 26.7 $\pm$ 9.1 a |
| Sabrina | 32.4 $\pm$ 3.4 a | 18.8 | 4.92 $\pm$ 1.1 a | 32.7 | 17.1 | 20.3 $\pm$ 9.8 a |
| LSD ($p < 0.05$) | $\pm$1.57 | - | $\pm$0.47 | - | - | $\pm$5.09 |
| Nutrient solution | | | | | | |
| S1 | 27.7 | 15.5 $^z$ $\pm$ 2.7 $^y$ b | 3.17 $^z$ $\pm$ 0.16 $^y$ c | 29.6 $^z$ $\pm$ 2.8 $^y$ ab | 16.1 $^z$ $\pm$ 1.8 $^y$ b | 18.8 $^z$ $\pm$ 6.4 $^y$ b |
| S2 | 27.1 | 20.0 $\pm$ 2.7 a | 4.92 $\pm$ 0.16 a | 30.2 $\pm$ 2.0 a | 19.4 $\pm$ 2.3 a | 26.7 $\pm$ 3.7 a |
| S3 | 26.7 | 18.1 $\pm$ 2.9 a | 3.74 $\pm$ 0.16 b | 28.3 $\pm$ 2.1 b | 17.2 $\pm$ 1.9 b | 20.3 $\pm$ 4.8 b |
| LSD ($p < 0.05$) | - | $\pm$2.30 | $\pm$0.47 | $\pm$1.57 | $\pm$1.32 | $\pm$5.09 |
| ANOVA (df) | | | | | | |
| Factor (df) | | | Percentage of the total sum of squares | | | |
| Cultivar (2) | 83.1 ** | 12.6 n.s. | 57.6 ** | 72.8 ** | 11.0 n.s. | 63.06 ** |
| Nutrient solution (2) | 0.9 n.s. | 40.6 ** | 27.7 ** | 4.6 n.s | 53.3 ** | 13.3 ** |
| Cultivar–nutrient solution (4) | 4.48 n.s. | 4.17 n.s. | 6.81 * | 8.93 * | 3.68 n.s. | 3.91 n.s. |

K:N balance: S1 (1.3/2.0), S2 (2.6/1.0), S3 (3.0/0.6) in growth and productive period. $^z$: mean value; $^y$: standard deviation. Values within a column with different letters differ at $p < 0.05$ using the LSD test. * and ** indicate significances at $p < 0.05$ and $p < 0.01$, respectively; n.s.: non-significant.

Figure 2 presents the double cultivar–nutrient solution interaction occurring in the early yield crops. The differences between nutrient solutions are only statistically significant ($p < 0.05$) in the Fortuna and Sabrina cultivars, which obtained the highest early yield figures for the plants irrigated with S2. Conversely, in the San Andreas cultivar there were no statistically significant differences between nutrient solutions S2 and S3, and those values were higher than the others ($p < 0.05$). Furthermore, statistically significant differences were observed between cultivars irrigated with S2 ($p < 0.05$), and the highest yields were obtained for plants of the Sabrina cultivar, followed by San Andreas.

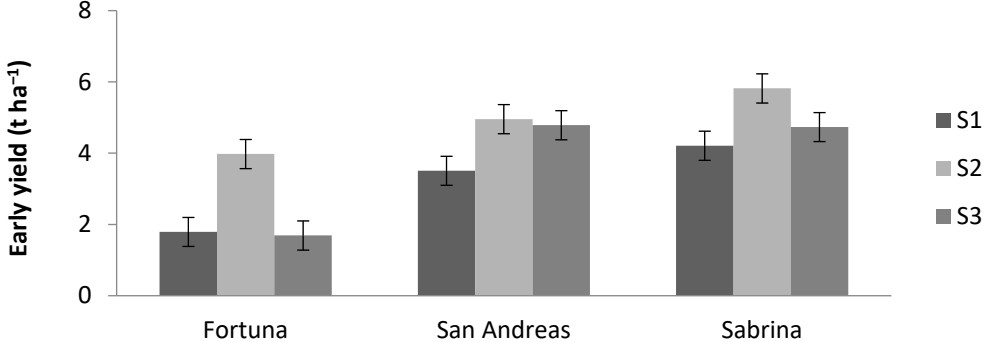

**Figure 2.** Mean values of early yield for three strawberry cultivars (Fortuna, San Andreas and Sabrina) irrigated with three nutrient solutions. K:N balance: S1 (1.3/2.0), S2 (2.6/1.0), S3 (3.0/0.6) in growth and productive period. Vertical bars indicate LSD value ($p < 0.05$).

Figure 3 presents the cultivar–nutrient solution interaction in relation to total production fruit size. The differences between nutrient solutions were statistically significant ($p < 0.05$) for Fortuna, which presented higher values for plants irrigated with S1 and S2, while for cultivars San Andreas and Sabrina, there were no statistically significant differences between the nutrient solutions. In addition, no statistically significant differences were observed between the cultivars irrigated using S1 and S2.

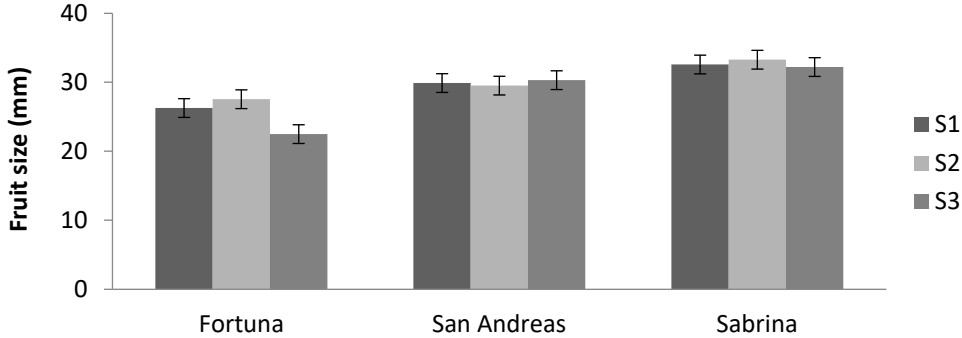

**Figure 3.** Mean values of fruit size of the total yield for three strawberry cultivars (Fortuna, San Andreas and Sabrina) irrigated with three nutrient solutions. K:N balance: S1 (1.3/2.0), S2 (2.6/1.0), S3 (3.0/0.6) in growth and productive period. Vertical bars indicate LSD value ($p < 0.05$).

Our trial results show that the potassium–nitrogen balance used in the fertilization program affects the fruit size and yield of strawberries. The improvement in strawberry growth parameters can be explained by the potassium effect, which positively and significantly influences the yield outcome [34,43]. Indeed, increased potassium levels in the nutrient solution stimulates the strawberry plant's growth by increasing carbohydrate synthesis activity due to the increased chlorophyll content [40], thus leading to improved fruiting dynamics [31,44]. A high potassium content in the fertigation solution enhances the average weight of strawberries [45] and the yield values of both tomatoes [46] and strawberries [40,47,48]. Lieten [49] showed that a low potassium concentration leads to a loss of plant vigor and a decrease in the yield. In fact, strawberry production is maximized via the use of potassium fertilization practices [50].

### 3.3. Effect of K:N Balance on the Quality Parameters of Strawberry Plants Grown in Soilless Conditions

In the present study, the variability of the factors studied varied from 0.10 to 76.03% of the total variability (Table 4). According to these results, the high and statistically significant influence ($p < 0.01$) of the cultivar factor remains salient out of all the other quality parameters, except for total soluble solids, which presented values ranging from 26.9 to 76.09%. On the other hand, the nutrient solution factor showed a statistically significant effect ($p < 0.01$) with variability values of 42.9 and 19.4% for total soluble solids and shelf-life, respectively. This greater statistical influence corresponds to the strong influence of the nutrient solution factor on taste ($p < 0.01$), since the variability explained by this factor presented a value of 39.3%. Regarding the double interactions that occurred, they were statistically significant ($p < 0.05$) only for the total soluble solids, presenting 23.8% of the total variability.

The differences present between the cultivars show that Sabrina fruits contained a higher acidity level, with statistically significant differences ($p < 0.05$), while the fruits of Fortuna and San Andreas showed a higher maturity index ($p < 0.05$), although the highest taste value was obtained for the fruits of the San Andreas cultivar. Finally, the dry matter content and shelf-life values were higher in the Sabrina variety, a result consistent with the higher acidity value. Therefore, fruits of the Sabrina cultivar have a better storage quality according to Havlin et al. [51].

**Table 4.** Mean values, standard error (±) and analysis of variance for quality parameters (total soluble solids, titratable acidity, maturity index, taste, shelf-life and dry matter content) of three strawberry cultivars (Fortuna, San Andreas and Sabrina) irrigated with three nutrient solutions.

| Factor | Total Soluble Solids (TSS) | Titratable Acidity (%) | Maturity Index | Taste (1–4) | Shelf-Life (% Weight Loss) | Dry Matter Content (%) |
|---|---|---|---|---|---|---|
| Cultivar | | | | | | |
| Fortuna | 6.49 | $0.64^{z} \pm 0.08^{y}$ c | $10.3^{z} \pm 1.9^{y}$ a | $2.20^{z} \pm 0.15^{y}$ b | $7.01^{z} \pm 1.23^{y}$ b | $7.24^{z} \pm 1.21^{y}$ b |
| San Andreas | 6.83 | $0.74 \pm 0.10$ b | $9.3 \pm 2.0$ a | $2.44 \pm 0.13$ a | $7.07 \pm 1.26$ b | $7.25 \pm 1.25$ b |
| Sabrina | 6.94 | $0.91 \pm 0.10$ a | $7.6 \pm 2.1$ b | $2.31 \pm 0.15$ ab | $4.01 \pm 1.27$ a | $8.81 \pm 1.26$ a |
| LSD ($p < 0.05$) | - | ±0.07 | ±1.27 | ±0.15 | ±0.86 | ±1.19 |
| Nutrient solution | | | | | | |
| S1 | $6.27^{z} \pm 0.65^{y}$ c | 0.76 | 8.3 | $2.22^{z} \pm 0.14^{y}$ b | $6.03^{z} \pm 1.80^{y}$ b | 7.09 |
| S2 | $7.29 \pm 0.60$ a | 0.76 | 9.8 | $2.47 \pm 0.14$ a | $5.05 \pm 1.90$ a | 8.29 |
| S3 | $6.71 \pm 0.60$ b | 0.77 | 9.1 | $2.27 \pm 0.16$ b | $7.01 \pm 1.62$ c | 7.89 |
| LSD ($p < 0.05$) | ±0.38 | - | - | ±0.15 | ±0.86 | - |
| ANOVA (df) | | | | | | |
| Factor (df) | | | Percentage of the total sum of squares | | | |
| Cultivar (2) | 9.23 [n.s.] | 76.03 ** | 45.4 ** | 26.9 ** | 61.8 ** | 30.6 * |
| Nutrient solution (2) | 42.9 ** | 0.10 [n.s.] | 12.1 [n.s.] | 39.3 ** | 19.4 ** | 14.1 [n.s.] |
| Cultivar–nutrient solution (4) | 23.8 * | 5.45 [n.s.] | 3.36 [n.s.] | 3.93 [n.s.] | 3.56 [n.s.] | 1.17 [n.s.] |

K:N balance: S1 (1.3/2.0), S2 (2.6/1.0), S3 (3.0/0.6) in growth and productive period. $^{z}$: mean value; $^{y}$: standard deviation. Values within a column with different letters differ at $p < 0.05$ using the LSD test. * and ** indicate significances at $p < 0.05$ and $p < 0.01$, respectively; [n.s.]: non-significant.

The nutrient solution with a high K:N balance during the growth period and a low balance during the fruit-production period (S2) significantly increased the total soluble solids (TSS) of the strawberry fruits (Table 4). The recorded values for the quality parameters reveal that the highest total soluble solids content was obtained for plants that received the S2 nutrient solution which was 7.29, and the lowest value was observed for S1 which represents the practice of strawberry growers (Table 4). However, the differences observed between the three solutions for acidity content were not statistically significant.

The analysis of variance revealed significant differences between the effects of the three solutions on the fruit's shelf-life. After 5 days of storage at 3 °C, fruits treated with solution S2 with a K:N balance of 2.6 during the growth period and 1.0 during the production period showed the best fruit shelf-life values presenting less weight loss (Table 4).

Table 4 shows that the S2 nutrient solution significantly increased the total soluble solids, taste and shelf-life parameters by 14%, 10% and 19%, respectively, compared to the practice of strawberry growers (S1).

Nutrient solution S2 also presented the highest values for the strawberries' dry matter content (Table 4). This nutrient solution with a high K:N balance during the growth period and low K:N balance during the fruit production period increased the dry matter content by 14.5% compared to the practice of strawberry growers. However, the differences observed between the three solutions were not statistically significant.

The sensory analysis we conducted showed that the fruits obtained from plants fertilized with the S2 nutrient solution were perceived as the sweetest and most preferred (Table 4).

The cultivar–nutrient solution interaction occurring for the total soluble solids showed no statistically significant differences between nutrient solutions in the San Andreas cultivar (Figure 4), while the Fortuna and Sabrina cultivars showed higher total soluble solids contents in the S2 and S3 solutions ($p < 0.05$), with the plants irrigated with S2 obtaining the highest total soluble solids values. In relation to the effect of each nutrient solution on each variety, it is evident that the highest total soluble solids value was achieved by the plants of the Sabrina cultivar treated with S2. Gündüz and Özbay [52] tested Fortuna, San Andreas and Sabrina cultivars with five other genotypes obtained from diverse strawberry-breeding programs and showed that the cultivars had significant effects on all the fruits' physical and chemical properties.

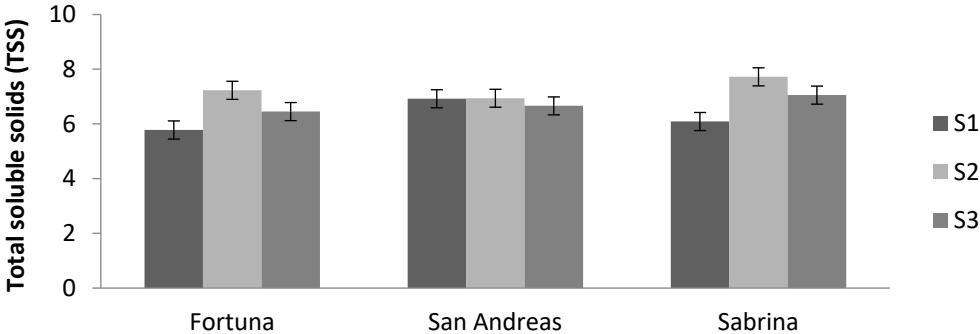

**Figure 4.** Mean values for total soluble solids for three strawberry cultivars (Fortuna, San Andreas and Sabrina) irrigated with three nutrient solutions. K:N balance: S1 (1.3/2.0), S2 (2.6/1.0), S3 (3.0/0.6) in growth and productive period. Vertical bars indicate LSD value ($p < 0.05$).

Our results show that the potassium–nitrogen balance used in the fertilization program affects the strawberries' quality parameters. Marschner [36] and Wang et al. [53] showed in their studies the major role that potassium plays in the processes of photosynthesis and carbohydrate metabolism. A high K:N balance improves the processes of carbon metabolism in functional leaves [54,55], and the optimal potassium supply allows for an improved total soluble solids content in reserve organs [56,57]. Shirko et al. [58] revealed that the nutrient solution supplied during the growth period affects the total soluble solids of the fruits. Indeed, a nutrient solution with high concentrations of potassium significantly increases the total soluble solids present in the strawberries [40,48]. As reported by Rodas et al. [9], the physicochemical properties of strawberries are influenced by combined doses of potassium and nitrogen applied by the fertigation process. According to other studies, increasing the potassium content compared to nitrogen in the nutrient solution increased the total soluble solids present in the strawberry fruits [40,47,48], which confirms the importance of potassium nutrition for the quality of strawberries.

The experiments prove that an adequate potassium intake level increases the dry matter content in strawberries [59]. Potassium is a general metabolic activator [60], improving synthesis of carbohydrates and producing new cells [31]. Morard and Raynal [61] showed that potassium consumption occurred almost simultaneously with dry matter production. It has been shown in the research that increasing potassium and nitrogen levels in nutrient solutions results in higher dry matter percentages in strawberries [62].

Asami et al. [63] observed that increasing nitrate concentration in the nutrient solution decreased the organoleptic quality of strawberries.

The stem of a perennial plant, called the crown, as in the case of strawberries, is an organ with the function of storing nutrients during winter and then transferring them to aid in the processes of plant growth and reproduction [5]. The strawberry plant stores potassium in its stem tissues during the growth period and reuses it during the high-demand phase that is the production period [64].

Another study conducted by Cakmak [57] proved that potassium is the most abundant of the cations present in the phloem (nearly 80%) aiding in the production and transportation of sugar to the reserve organs.

Therefore, an optimal potassium intake level encourages improved strawberry growth, productivity and quality parameters, which confirms the importance of potassium nutrition management in the field of strawberry nutrition.

## 4. Conclusions

Adequate potassium nutrition plays an important role in strawberry growth, productivity and quality parameters for all varieties. Based on the results of our analysis, we can conclude that nutrient solutions with a high K:N balance used during the growth period and low balance used during the fruit-production period (2.6/1.0) significantly increases the yield and improves the quality of strawberries grown in soilless conditions.



This nutrient solution consisting of a K:N balance, unlike farmers' current practice (1.3/2.0), increases the chlorophyll index by 8%, yield by 30% (7.9 t ha$^{-1}$), total soluble solids and dry matter content by 14%, and 15%, respectively, and improves both the taste and fruit shelf-life factors by 10%, and 19%, respectively.

Our research results confirm that an optimal potassium–nitrogen balance allows for better growth, productivity and quality of all strawberry cultivars grown in soilless conditions.

**Author Contributions:** Conceptualization, A.N. and A.B.; data curation, A.N., A.B. and H.B.; formal analysis, A.N., A.B., A.S.B. and H.B.; investigation, A.N., A.B., L.G. and H.B.; methodology, A.N., A.B. and A.S.B.; software, A.N., A.B. and A.S.B.; supervision, A.B., A.S.B. and L.G.; validation, A.N., A.B., A.S.B. and L.G.; visualization, A.N., A.B. and H.B.; writing—original draft preparation, A.N.; writing—review and editing, A.N., A.B., A.S.B. and L.G. All authors have read and agreed to the published version of the manuscript.

**Funding:** This research received no external funding.

**Conflicts of Interest:** The authors declare no conflict of interest.

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
