# Peer review of "The Effect of Potassium–Nitrogen Balance on the Yield and Quality of Strawberries Grown under Soilless Conditions"

_horticulturae, doi:10.3390/horticulturae9030304_

Round 1

Reviewer 1 Report

The paper deals with an interesting aspect of soilless strawberry cultivation for yield and quality aspects in relation to Nand K supply. There are some issues in the scientific language writing and presentation of the results. Author must revise the presentation style from present to past tense starting from the very first sentence of Abstract...the ethe purpose of this research was(is)....also first statement in Results and discussions as an example...please revise throughout the MS. Secondly, add standard deviations in the figures and significance.

The title of tables and figures need to be revised in due consideration of language and contents presented in them.

Use total soluble solids for TSS in place of sugar.

Chlorophyll index not content index.

I recommend to change figure presentation in a way to compare different cultivars within each of n-knbalance treatments (s1,s2,s3).

Overall, this ms required a significant improvement before go for publication. 

Author Response

Response to the reviewer 1 :

  • The paper deals with an interesting aspect of soilless strawberry cultivation for yield and quality aspects in relation to Nand K supply. There are some issues in the scientific language writing and presentation of the results. Author must revise the presentation style from present to past tense starting from the very first sentence of Abstract...the ethe purpose of this research was(is)....also first statement in Results and discussions as an example...please revise throughout the MS. Secondly, add standard deviations in the figures and significance.

verbs in the present tense have been changed to the past tense throughout the manuscript

  • The title of tables and figures need to be revised in due consideration of language and contents presented in them.

Title Table 1 has been changed to Nutrient solutions composition (mmol L-1).

 Title Table 2 has been changed to Mean values, standard error (±) and analysis of variance of Chlorophyll Index (CI) and  Stomatal conductance (s cm-1) of three strawberry cultivars (Fortuna, San Andreas and Sabrina) irrigated with three nutrient solutions

Title Figure 1 has been changed to Mean values of chlorophyll index on three strawberry cultivars (Fortuna, San Andreas and Sabrina) irrigated with three nutrient solutions

Title Table 3 has been changed to Mean values, standard error (±)  and analysis of variance of Productivity parameters (fruit size, weight fruit and yield) on early (March 31th) and total production (June 30th) of three strawberry cultivars (Fortuna, San Andreas and Sabrina) irrigated with three nutrient solutions

Title Figure 2 has been changed to Mean values of early yield on three strawberry cultivars (Fortuna, San Andreas and Sabrina) irrigated with three nutrient solutions

Title Figure 3 has been changed to Mean values on fruit size of the total yield for three strawberry cultivars (Fortuna, San Andreas and Sabrina) irrigated with three nutrient solutions

Title Table 4 has been changed to Mean values, standard error (±)  and analysis of variance of quality parameters (total soluble solids, titratable acidity, maturity index, taste, shelf-life and dry matter content) of three strawberry cultivars (Fortuna, San Andreas and Sabrina) irrigated with three nutrient solutions

Title Figure 4 has been changed to Mean values on fruits total soluble solids for three strawberry cultivars (Fortuna, San Andreas and Sabrina) irrigated with three nutrient solutions

  • Use total soluble solids for TSS in place of sugar.

Sugar content has been changed to total soluble solids

  • Chlorophyll index not content index.

chlorophyll content index has been changed to Chlorophyll index

  • I recommend to change figure presentation in a way to compare different cultivars within each of n-knbalance treatments (s1,s2,s3).

We have not changed the figures because the double interaction can be interpreted in both directions and the value of the LSD interval is the same for the interpretation of the interaction. Moreover, the cultivar factor has shown a high variability with respect to the total as shown in the analysis of variance, and the value of the Snedecor F-test is statically significant (p<0.01). For these two reasons we consider the graphs to be correct.  Other reviewers do not consider this change necessary.

  • Overall, this ms required a significant improvement before go for publication.

All comments have been inclued to improve the manuscript

Reviewer 2 Report

The manuscript is interesting, the obtained data do not match the generally accepted knowledge about the necessary K/N ratio for plants. It is interesting, but the result could be influenced by the growing conditions, so to draw conclusions, at least two seasons of research would be necessary.

I think the numbers need to be improved. Be sure to review the numbers in the last column of Table 3. Review the references as some sources do not have page numbers and 1st, 27th reference should be written as a footnote or web page.

Other comments are in the attached manuscript.

Author Response

Response to the reviewer 2 :

  • P1, Line 14

 Deleted

  • P1, Line 25 Keywords repeat the title

the new keywords are the following: Fertigation, nutrient solution, ratio, productivity, total solube solids, maturity index 

  • P1, Line 29 Unsusual classification. If there are yellow and red raspberries, do only the yellow ones count here ?

Red berry has been changed to berry

  • P2, Line 84

ml has been changed to mL

  • P3, Line 137

80ºC has been changed to 80°C

  • P5, Fig. 1 The figure would be easier to perceive if the figure border will be removed and the error bars will be at each bar ; I can see

The figure border has been removed.

The error bar is the LSD range (p<0.05) and the value is only one for all treatments

  • P6, Table 3 Please check the numbers

The numbers will checked

  • P7, Fig. 2 the same comments as fig.1

The figure border has been removed.

The error bar is the LSD range (p<0.05) and the value is only one for all treatments

t ha-1 has been changed to t ha-1

  • P9, Fig. 4 the same comments as fig.1

The figure border has been removed.

The error bar is the LSD range (p<0.05) and the value is only one for all treatments

  • P9, Line 336 Reference is about solanaceoys plants !

The reference has been changed to [57] Kaya, C.; Higgs, D.; Saltali, K.; Gezerel, O. Response of strawberry grown at high salinity and alkalinity to supplementary potassium. Journal of Plant Nutrition, 2002, 25(7) : 1415–1427. https://doi.org/10.1081/PLN-120005399

Reviewer 3 Report

Thanks for the opportunity to evaluate this document. The study investigated the effect of potassium-nitrogen balance on the production and quality parameters of strawberries grown above ground by testing three fertilization programs to optimize the efficiency of the use of nitrogen and potassium fertilizers in the two important phenological phases of the crop through the management of potassium sulfate in the nutrient solution. The manuscript is well-written and scientifically sound. The required changes should focus on better connecting the justification, and improving the grammar and style. Additionally, authors should reduce the repetitive description of results and the text explaining other studies and focus more on explaining why they think their study obtained such results. That way, we’ll benefit from a direct contribution to this study instead of data compiled from other trials. I made several suggestions directly in the text, and I hope they help the authors tell a nice story.

Author Response

Response to the reviewer 3 :

  • P1, Line 28 The text is lacking fluence and connectivity among topics and paragraphs

The text has been revised.

  • P1, Line 44 Please use soluble solids content instead of Brix

Brix has been changed to total soluble solids according to the two others reviewers

  • P2, Line 58 Move to the end of this sentence

The sentence has been moved to the end line 63

  • P2, Line 81 Sand is a soil. Please remove the mention of soilless substrate in the text

We have considered soilless substrate term according to Savvas and Passam (2002). We have incluied this reference in the text. Sand is considered an inorganic subtrate in this reference.

  • P3, Line 105 How did you come up with these values ? Please provide more info in the introduction

We have considered S1 control nutrient solution accordding to [28] and S2 is doble ratio S1 durign the growth period and half ratio S1 in the production period. Finally, we have considered that S3 is modified S2 and we propose to reduce ratio 0.6 in production period

  • P3, Line 132 Excellent approach !

Thank you very much

  • P4, Line 139 the word obtained will be removed

The word obtained has been removed

  • P4, Line 145 This is not the correct place for the objectives

The paragraph has been removed

  • P7, Fig. 2 Not sur what the authors mean by this image over the bars

We have removed LSD en the legend

  • P7, Line 258 This is a well-know reason. What the authors think it happened ?

These results are according to references cited

  • P8, Line 282 Authors focus much more on presenting the results than discussing it, not explaining why they obtained such results

We have incluied a discussion about this result : So, fruits of Sabrina have a better storage quality according to Havlin et al. (2014).     

  • P8, Table 4 Convert commas to dots

The commas will changed to dots

Reviewer 4 Report

The study is well structured and many parameters are examined. But the written language is very difficult to understand. Some terms are used incorrectly. The language needs to be revised. There are also statistical errors in tables and graphs.  My criticisms and comments are below.

Line 15  add .   and delete The  .. English is not understood

Line 18 rewrite

Line 19-21 rewrite For a better understanding of this part, also mention the rates during the growth period.

It would be more appropriate to write S1-S2-S3 for the growth period and S4-S5-S6 for the fruiting period both in the summary and in the text, in the tables and graphics. (this numbering is necessary because the doses are different). Should be corrected in the whole article.

Line 23-24 do not repeat respectively

Line 47 delete strawberry, delete "namely" and  write "such as"

Line 58-63 merge paragraph

Line 77-78 can not understand

Line 119 delete the productive parameters and write pomological analyses

Line 133 delete by a dozen and add give the number or person

Write a detailed description about self life.

Line 146 grown above ground? What is it meant? soilless culture

Line 148-149 potassium sulfate ?

in all tables and graphics;

In cases where the interaction is significant, it is not appropriate to interpret the main effects one by one. This should be noted and corrected in tulle tables.

Standard errors and lsd values should be added for important parameters in both tables and graphs.

lsd parts in graphics are placed as meaningless. lsd line should be removed instead of error bars and lettering should be added.

Since interactions are given in the graphics, it is very important to add lettering in these parts.

delete all below the graphics "the vertical bar indicates..... "  sentence

Line 205 can not understand

Line 206 delete double

Line 247 delete dual

Line 309 delete double

Author Response

Respnse to the reviewer 4 :

  • P1, Line 11  add .   and delete The  .. English is not understood

The dot has been added and The has been removed

  • P1, Line 18 rewrite

The sentence has been changed to : The experimental design was a randomized complete block with three replications.

  • P1, Line 19-21 rewrite For a better understanding of this part, also mention the rates during the growth period.

The paragraph has been changed to : Findings showed that strawberry plants receiving a nutrient solution with K:N balance high during the growth period and low during the production period were recorded the higther growth and fruit parameters.

  • It would be more appropriate to write S1-S2-S3 for the growth period and S4-S5-S6 for the fruiting period both in the summary and in the text, in the tables and graphics. (this numbering is necessary because the doses are different). Should be corrected in the whole article.

The same plants were irrigated with S1/S4, S2/S5 and S3/S6. So, there are only three solutions. But, the nutrient solution composition changed from growth period to production period.

  • P1, Line 23-24 do not repeat respectively

The repetation has been removed

  • P2, Line 47 delete strawberry, delete "namely" and  write "such as"

Comments correspond to Line 51

The sentence has been changed to : parameters such as total soluble solids, titratable acidity and fruit juice pH are…

  • P2, Line 58-63 merge paragraph

The sentence has been moved to the end line 63

  • P2, Line 77-78 can not understand

The paragraph has been changed to : A pot experiment was conducted at greenhouse of the Hassan II Agronomic and Veterinary Institute in Rabat, Morocco (33°58’43,00’’N 6°51’50,75’’O; Altitude: 127 m).

  • P3, Line 119 delete the productive parameters and write pomological analyses

The sentence has been changed to : the pomological analyses performed were fruiting dynamic recorded as number of ….

  • P3, Line 133 delete by a dozen and add give the number or person

The sentence has been changed to : ….. by 12 people.

  • Write a detailed description about self life.

The description about shelf life is mentioned in Line 133-135.

  • P4, Line 146 grown above ground? What is it meant? soilless culture

The paragraph has been removed as requested by the reviewer 3

  • P4, Line 148-149 potassium sulfate ?

The paragraph has been removed as requested by the reviewer 3

  • in all tables and graphics;
  • In cases where the interaction is significant, it is not appropriate to interpret the main effects one by one. This should be noted and corrected in tulle tables.

The interpretation of simple effects and interaction has been corrected.Standard errors and lsd values should be added for important parameters in both tables and graphs.

The values have been added.

  • lsd parts in graphics are placed as meaningless. lsd line should be removed instead of error bars and lettering should be added.

The LSD interval (p<0.05) has been inserted in each bar to better interpret the interaction. The interpretation of the double interaction presents a double sense of explanation (with double letters), for this reason it has been considered more appropriate to include the LSD interval in each treatment for better interpretation (as has been done in other manuscripts).Since interactions are given in the graphics, it is very important to add lettering in these parts.

We consider that the above answer can be accepted in this question as well.

  • Delete all below the graphics "the vertical bar indicates..... "  sentence

the sentence "the vertical bar indicates..... "   has been removed

  • P5, Line 205 can not understand

We calculated the sum of squares of each factor and the total sum of squares in the analysis of variance. Then, the sum of squares of a factor was divided by the total sum of squares, and its percentage was calculated. This value explains the variability due to the factor studied with respect to the total variability of the experiment. A high value of the variability of the factor indicates that its statistic influence is greater than other factors considered in this experiment or not considered. The sum of the variability of the variety factor, the nutrient solution and their interaction results in values higher than 80% in many of the parameters analyzed, and therefore only 20% of the variability is due to other factors not considered in our experiment (other than the variety and the nutrient solution).

  • P5, Line 206 delete double

It has been delete

  • P7, Line 247 delete dual

Dual has been removed

Round 2

Reviewer 1 Report

The authors have revised the manuscript considering most of the comments/ suggestions, on which they do not agree have provided explanations.

I still find some minor errors e.g., using sugar in the Abstract and Conclusion section instead of using TSS (total soluble solids).

L 353: correct 'storinge' 

L 356-357: 'we could be can conclude' needs to be corrected. 

Insert the full form of S1, S2 and S3 in the footnote or caption of each table and figure.

It is better to add Standard Deviation after mean values instead of SE which has the same value for each factor for each parameter.

 Some correction in the formatting of references is needed e.g., a comma after volume (issue), Abbreviation for Journal names, etc.

Author Response

  • I still find some minor errors e.g., using sugar in the Abstract and Conclusion section instead of using TSS (total soluble solids).

Sugar has been changed to total soluble solids in the Abstract and Conclusion section.

  • L 353: correct 'storinge'

Storinge has been changed to storing

  • L 356-357: 'we could be can conclude' needs to be corrected.

We could be can conclude  has been changed to  we can conclude

  • Insert the full form of S1, S2 and S3 in the footnote or caption of each table and figure.

The full form of S1, S2 and S3 have been inculed in the caption of each table and figure

  • It is better to add Standard Deviation after mean values instead of SE which has the same Value for each factor for each parameter.

The Standard Deviation has been added for each factor and parameter

  • Some correction in the formatting of references is needed e.g., a comma after volume (issue), Abbreviation for Journal names, etc.

The rectifications of references have been made.

Reviewer 4 Report

The authors have made the desired arrangements and it is appropriate to publish it as it is. I wish you success in your work.

Author Response

We would like to thank them for all their work and suggestions for improving the manuscript.